# How to Manage a Patient with Haemophilia and ACS Requiring PCI: A Battle between Bleeding and Thrombosis

**DOI:** 10.3390/medicina57040352

**Published:** 2021-04-07

**Authors:** Konstantinos C. Theodoropoulos, Sofia Vakalopoulou, Maria Oikonomou, George Stavropoulos, Antonios Ziakas, Ioannis Kanonidis, George Kassimis

**Affiliations:** 1Second Cardiology Department, Hippokration Hospital, Aristotle University of Thessaloniki, 54642 Thessaloniki, Greece; ktheod2005@hotmail.com (K.C.T.); mariaoikonomou148@gmail.com (M.O.); stavropoulosgeo@gmail.com (G.S.); bkkaut@med.auth.gr (I.K.); 2First Cardiology Department, University General Hospital AHEPA, Aristotle University of Thessaloniki, 54642 Thessaloniki, Greece; tonyziakas@hotmail.com; 3Haemophilia Centre of Northern Greece, Second Propaedeutic Department of Internal Medicine, Medical School, Hippokration Hospital, Aristotle University of Thessaloniki, 54642 Thessaloniki, Greece; svakalopoulou@yahoo.com

**Keywords:** Haemophilia B, factor IX, coagulation factors, bleeding, thrombosis, PCI, NSTE-ACS

## Abstract

We present the case of a 70-year-old man with a history of haemophilia B, who presented to our hospital with a non-ST-elevation myocardial infarction. The patient, following consultation by a haemophilia expert, was revascularized with percutaneous coronary intervention (PCI) under adequate clotting factor administration. Patients with haemophilia and acute coronary syndrome, are susceptible to periprocedural bleeding and thrombotic events during PCI, and therefore a balanced management plan should always be implemented by a multidisciplinary team.

## 1. Introduction

Coronary artery disease (CAD) is the most common type of cardiac disease with the highest mortality globally [1]. Non-ST-elevation acute coronary syndromes (NSTE-ACS) represent 70% of all ACSs, while patients with NSTE-ACS tend to be older and have more comorbidities than patients with ST-elevation myocardial infarction (STEMI) [2]. Invasive strategy with a coronary angiogram and possible percutaneous coronary intervention (PCI) along with dual antiplatelet therapy (DAPT) and anticoagulation is the standard of care for patients with NSTE-ACS according to current guidelines [2,3]. However, periprocedural bleeding is one of the most frequent complications of PCI and is associated with an increased risk of both short- and long-term mortality [4]. A high incidence of bleeding events occurs in patients with various haematological disorders such as anaemia, thrombocytopenia, or haemophilia who undergo PCI.

Haemophilia A (HA) and haemophilia B (HB) are recessive, X-linked inherited blood disorders caused by a deficiency of coagulation factor VIII (FVIII) and factor IX (FIX) respectively [5]. They affect mainly males and their clinical manifestations are similar, however it is generally considered that bleeding episodes in patients with HA are more severe and occur at higher frequency than in patients with HB [5]. HB was first reported in 1952 and was called “Christmas disease” after the name of the patient, a 10-year-old boy named Stephen Christmas [6]. The severity of haemophilia is inversely related to the level of functional FVIII or FIX present and is categorized into mild, moderate, and severe disease, as follows: severe (factor level < 1% of normal), moderate (factor level 1–5% of normal) and mild (factor level > 5% and <40% of normal) [7]. However, current stratification based on factor VII or IX levels is not always reflecting the clinical severity of the disease. Haemophilia patients are usually well known to their haemophilia centres, which often have specialized treatment protocols. Bleeding episodes can range from minor/moderate (haemarthrosis, epistaxis) to severe (intracerebral bleed or limp muscle bleeds leading to compartment syndrome). Treatment of haemophilia has evolved dramatically in the last decades, transforming haemophilia from an often-fatal hereditary bleeding disease to one with available, safe and effective, treatment. Haemophilia treatment is based on missing factor replacement with FVIII or FIX concentrates. These concentrates can be used as either prophylaxis or treatment of bleeding episodes [8]. 

## 2. Case Presentation

A 70-year-old man developed sudden onset severe central chest pain lasting in total 2 h. Cardiovascular risk factors included only hypertension and the patient had a history of mild haemophilia B and chronic obstructive pulmonary disease (COPD). He presented to our institute some hours later, where a 12-lead electrocardiogram demonstrated T wave inversion in leads III and V1-V6 (Figure 1) and hs-TnI was mildly elevated (120 ng/L, normal < 14 ng/L), therefore a NSTE-ACS was diagnosed. Blood pressure was 160/90 mmHg. Baseline Factor IX activity was 24.9% (Figure 2), activated Partial Thromboplastin Time [aPTT] was 37.9 s and international normalized ratio (INR) 0.98 (INR is normal and aPTT is prolonged in haemophilia patients). Haemogram and biochemical parameters were all within normal ranges. 

As we adopted an early intervention strategy, the patient was referred to haematology for consultation and joined management. The patient was loaded with dual antiplatelet therapy (aspirin and clopidogrel), b-blocker and lansoprazole. The coronary angiogram was performed the next day and periprocedural intravenous unfractionated heparin (UFH) was used, guided by the ACT (activated clotting time) levels. Although ACT is prolonged in patients with haemophilia, replacement of the deficient FVIII or FIX suppress this phenomenon. Despite this, ACT remains a useful point-of-care test in order to monitor coagulation during PCI and to guide unfractionated heparin (UFH) treatment. Recombinant factor IX (with a periprocedural level target of >80%) was also administered (Figure 2). Given the baseline level (24.9%), the patient received 5000 units of recombinant factor IX (Benefix, Pfizer, Philadelelphia, PA, USA) intravenously two hours before coronary angiography (CAG). CAG was performed using the left radial artery access and demonstrated unobstructed right and left circumflex coronary arteries. The left anterior descending artery (LAD) had proximally a severe non-calcified lesion causing 99% stenosis (Figure 3A). The culprit lesion was treated with a 2.75 mm × 18 mm zotarolimus eluting stent and post-dilated with a non-compliant balloon 3.0 mm × 15 mm at 18 atm with a good final angiographic result (Figure 3B). Three hours later, the TR BAND^®^ Radial Compression Device was removed and neither bleeding nor haematoma was observed. The patient had an uneventful recovery and was discharged the following day with recommendation for DAPT (aspirin and clopidogrel) for 1 month. He continued to receive prophylaxis therapy with regular administration of FIX concentrates to keep trough levels around ≥30% while on DAPT (Figure 2). He was completely asymptomatic in a follow-up appointment one month later, with preserved left ventricular systolic function on transthoracic echocardiography. 

## 3. Discussion

### 3.1. Haemophilia and Coronary Artery Disease

Haemophilia is an inherited blood disorder due to deficiency of coagulation factors VIII or IX. Patients with haemophilia suffer from minor or major bleeding events. Although haemophilia patients appear to have lower mortality due to ischemic heart disease compared to the general population, the prevalence of cardiovascular risk factors in haemophilia patients is similar to the prevalence in the general population, and hypertension is even more common [9,10]. Lower mortality of haemophiliacs due to ischaemic heart disease can be attributed to their hypocoagulable state and reduced thrombus formation [9,10]. Nonetheless, haemophiliacs have the same degree of atherosclerosis as the general population and the incidence of CAD in haemophilia patients is increasing, as life expectancy of these patients approaches nowadays that of the general population [9,10].

### 3.2. Haemophilia and Acute Coronary Syndromes

Haemophiliacs carry an increased risk for periprocedural bleeding and PCI is quite challenging for the interventional cardiologist when attempted, especially if haemophilia is severe. Nevertheless, there are no dedicated guidelines for the management of ACS in patients with haemophilia as patients with bleeding disorders have generally been excluded from ACS clinical trials. Current practice is supported only by small retrospective studies or consensus statements and consultation by haematologists is always necessary for the management of these patients.

Patients with haemophilia normally receive replacement therapy with factor concentrates (FVIII or FIX) either as regular prophylaxis in severe haemophilia or as treatment on demand in case of bleeding episodes. Ideally, patients with haemophilia should carry an identifiable information card that, in case of presentation with ACS, would alert treating physicians to the presence of a bleeding disorder, hence a haemophilia expert should be consulted as soon as possible [11]. 

Early mechanical revascularization with PCI is the treatment of choice for patients with an ACS (STEMI or high risk NSTE-ACS) as per European Society of Cardiology guidelines [3,12]. In patients with haemophilia presenting with STEMI, fibrinolysis could be considered as a treatment option if primary PCI is not available, but always after the replacement therapy of the clotting factor [11]. In patients with haemophilia who present with high-risk ACS and need early revascularisation, PCI should be performed promptly as in general population. However, depending on the severity of the bleeding disorder, the urgency of the procedure and the availability of clotting factor concentrates, replacement therapy supervised by a haemophilia specialist should be initiated as soon as possible, prior the procedure or in parallel to the procedure, but always before sheath removal [11]. The peak level of the impaired clotting factor should ideally exceed 80% of normal during the periprocedural period [11]. These manipulations are quite sensitive considering that infusion of factor concentrates itself has been associated with the development of ACS due to their pro-thrombotic properties [13].

### 3.3. Selection of Access Site

To improve procedural safety, angiography using a radial rather than femoral approach is recommended [11]. Radial approach makes compression easier and lowers the bleeding risk that femoral approach encounters (retroperitoneal haematoma). 

### 3.4. Selection of Stent

Another debate area in the management of patients with haemophilia and ACS requiring PCI is the type of stent that should be implanted. Although patients treated with drug-eluting stents (DES) have a lower risk of stent thrombosis and repeat revascularization compared to patients that receive bare metal stents (BMS), they require a longer treatment period with antiplatelet agents. Even though the use of BMS is clearly favoured by the expert consensus document written by the ADVANCE Working Group, the expert panel recognizes that DES should be considered in patients with prior restenosis or elevated risk of restenosis due to diabetes [11]. However, recent data suggests that use of a Zotarolimus-eluting stent or polymer-free Biolimus-coated stent, in high bleeding risk patients, when used with only 1 month of dual antiplatelet therapy, was proven superior to a BMS with respect to the primary safety and efficacy end points [14].

The ZEUS (Zotarolimus-Eluting Endeavor Sprint Stent in Uncertain DES Candidates) and SENIOR (Short Duration of Dual Antiplatelet Therapy with Synergy II Stent in Patients Older Than 75 Years Undergoing Percutaneous Coronary Revascularization) trials showed that such patients receiving the Endeavor zotarolimus-eluting stent or the bioresorbable polymer-based everolimus-eluting stent with abbreviated dual antiplatelet therapy (1 to 6 months) had better outcomes than patients receiving bare-metal stents [15,16]. The LEADERS FREE trial showed that polymer-free umirolimus coated stents were superior to bare-metal stents in terms of both safety and effectiveness in patients at high bleeding risk who were treated with 1 month of dual antiplatelet therapy [17]. Most recently, among patients at high bleeding risk who received 1-month DAPT after PCI, the use of polymer-based zotarolimus-eluting stents (ZES) (Resolute Onyx stent) was noninferior to the polymer-free drug-coated stents (BioFreedom stent) with regard to safety and effectiveness composite outcomes [18]. In conclusion, in the current era of PCI with the use of contemporary DES, refined PCI techniques and adjunctive pharmacotherapy, a strategy of a combination of a DES to reduce the risk of subsequent repeat revascularisations with a short BMS-like DAPT regimen to reduce the risk of a bleeding is an effective and safe option for haemophiliacs who undergo PCI. 

### 3.5. Selection of Anticoagulant Therapy

The use of anticoagulants in patients with haemophilia without factor replacement therapy is not recommended, while after factor replacement therapy anticoagulants can be given depending on thrombotic risk of the patient [11]. All range of anticoagulants can be theoretically used to treat haemophiliac patients with ACS, however there is a preference for UFH based on the short half-life and the fact that its anticoagulation effect can be easily assessed in the cath-lab with the ACT and reversed by protamine administration [11].

### 3.6. Selection of Antiplatelet Therapy

Dual antiplatelet therapy (DAPT) consisting of aspirin and a P2Y_12_ inhibitor, is recommended post-PCI to prevent stent thrombosis [3,12]. Since haemophilia is not associated with any platelet abnormality, antiplatelet agents are given in the setting of an ACS as in general population. There is currently no clear recommendation regarding which P2Y_12_ inhibitor is better in this clinical setting. However, in most of the cases reported, aspirin along with clopidogrel was used for DAPT. The addition of proton pump inhibitors is recommended for haemophiliacs on antiplatelet therapy to reduce the bleeding risk [11]. As in general population, the use of glycoprotein IIb/IIIa inhibitors should be considered only for selected high-risk situations (bail-out therapy) and always following administration of replacement clotting factor therapy [11].

### 3.7. Management of Bleeding Risk Associated with Antiplatelet Therapy

The consensus document by the ADVANCE Working Groups suggests that the administration of antiplatelet agents should not be delayed in the acute setting, however the expert panel recommends that clotting factor levels should be around 50% 24 h post PCI and minimum trough clotting factor levels should not fall below 5–15% when patient is on DAPT or below 1% when on aspirin alone [11]. However, more recent data suggests that clotting factor levels should be around 80% 24 h post PCI and minimum trough clotting factor levels ≥ 20–30% when the patient is on DAPT [19,20,21] leading to the least amount of bleeding. The duration of dual antiplatelet therapy post-PCI should be reduced to a minimum, ideally 1 to 6 months depending on the type of stent [11]. Close follow up, either in the clinic or remotely via phone calls is considered necessary for these high bleeding risk patients to monitor for bleeding events.

## 4. Conclusions

Patients with haemophilia presenting with an ACS should be managed promptly by a multidisciplinary team that includes a haematologist. There is paucity of data on this field as our current practice is supported mainly by few expert consensus documents and retrospective studies. Revascularisation with PCI should be attempted, in patients with high-risk ACS, in a timely manner as in the general population just after or in parallel to the clotting factor replacement therapy. 

## Figures and Tables

**Figure 1 medicina-57-00352-f001:**
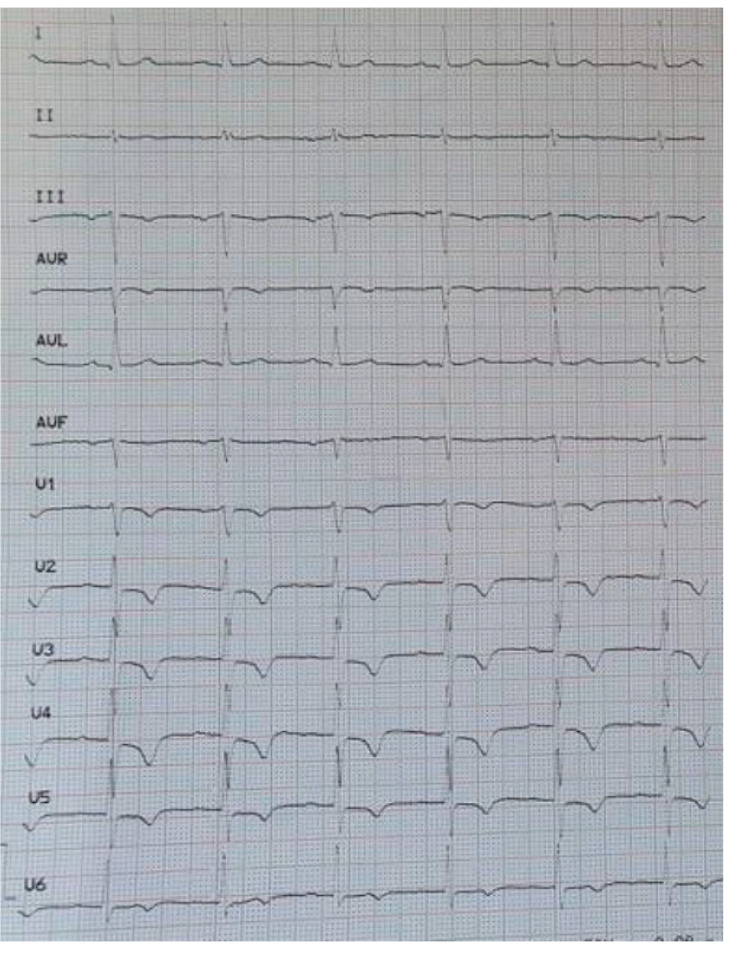
12 lead electrocardiogram (ECG) showing T wave inversion in leads III and V1–V6.

**Figure 2 medicina-57-00352-f002:**
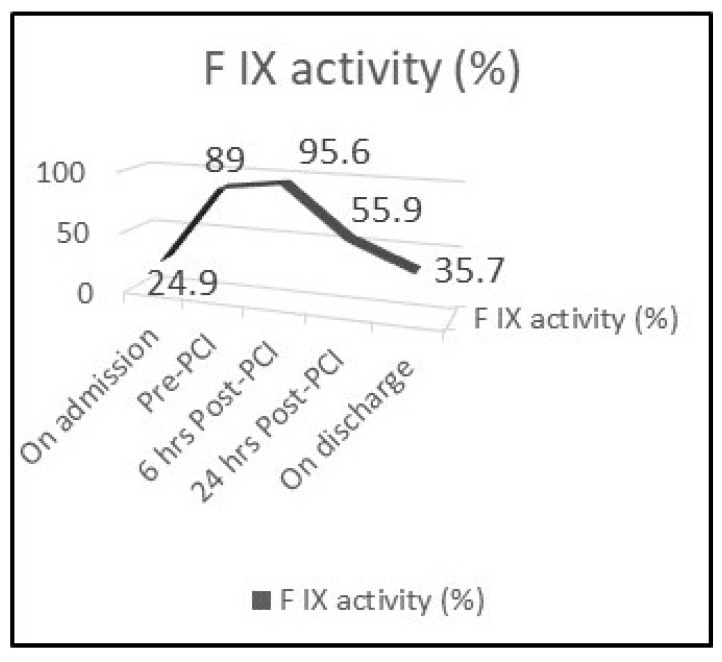
Plasma Factor IX activity before, during and after percutaneous coronary intervention (PCI).

**Figure 3 medicina-57-00352-f003:**
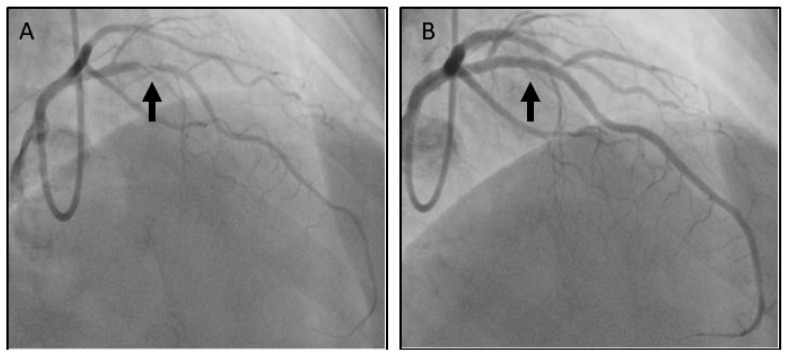
(**A**) Coronary angiogram showing severe proximal left anterior descending (LAD) lesion. (**B**) Coronary angiogram showing good angiographic result post-PCI.

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
