# Peer review of "How to Manage a Patient with Haemophilia and ACS Requiring PCI: A Battle between Bleeding and Thrombosis"

_medicina, 2021, doi:10.3390/medicina57040352_

Round 1
Reviewer 1 Report
This is an interesting case report, which touches on an important and emerging issue as persons with hemophilia get older. There are some grammatical issues in the paper, but I have more concerns with some of the content of the introduction and the discussion.
-After presenting the hemophilias, the focus is on HB, presumably because of the case. However, the information is relevant to both A and B, so it would be good to maintain that broad focus.
-The severity as stratified by factor level is current practice, but there is increasing recognition that the clinical phenotype does not always match the level. Stating that moderate patients bleed only after minor trauma and severe after major trauma overlooks those moderates who have more severe bleeding. This should not be stated so categorically here.
-The figure of the case report has four images which are not related. The ECG and factor level graph should stand on their own.
-The discussions of hemophilia and bleeding (3.1, 3.5, 3.6) are drawn primarily from a single consensus guideline. While there may not be much more available, this should be stated explicitly, with an acknowledgement of the limitations of such a narrow data set.
-The discussion of stent selection alludes to a study in patients with high bleeding risk. How many of those patients had hemophilia? Bleeding from other risk factors may be a very different consideration.
-In the case, the ACT was used to monitor periprocedural coagulation. The ACT will be impacted by factor levels. While I'm sure that was controlled for in the factor correction, it would be reasonable to mention the impact of factor deficiencies on the ACT as well as the aPTT.
Reviewer 2 Report
In the care report. Theodoropoulos C et al. presented a successful management of hemophilia B with a procedure of percutaneous coronary intervention. The report appears interesting, nonetheless, given no introduction of epidemiology of CAD in hemophilia patients, the significance of the manuscript is not fully justified.
Some other comments:
- The two parts in the “introduction” appeared totally isolated without a bridge.
- Not sure the significance of some citations, e.g.
Line 33: Ref 5-6: A high incidence of bleeding events is associated with various haematological disorders such as anaemia, thrombocy[1]35 topenia, or other coagulopathies [5,6].
Line 55: Ref 11: Desmopressin is an effective haemostatic agent that can …….
Round 2
Reviewer 1 Report
I appreciate the thoughtful response to the comments made previously, and the revision is clearly improved. However, I still have some concerns and suggestions related to my previous comments:
-The amendment made to clarify the stratification of severity is very awkwardly written and vague. It is oversimplification to say that moderates bleed 'most of the time after minor trauma' while milds bleed 'most of the time after major trauma'. The important concept to convey is that current stratification on factor level is not always reflected in clinical severity, and it would be better to make that statement more clearly than to try to explain each level of factor-derived severity.
-Comments have been added in parentheses that the ACT and aPTT are prolonged in hemophilia. This does not address the impact of hemophilia on the utility of point-of-care ACT monitoring in PCI, which is an important element to consider during the procedure. Of course, factor replacement can mitigate the variation, which could be included in the discussion.
-The figure is still a single figure - only the legend has been revised and separated, and no longer matches the image. The picture should be specifically divided into the three figures mentioned, each with their own legend.
-I appreciate the response to my comments about the limited data source available, and concur that there is not better evidence. In addition to the comment in section 3.2, it would help to reiterate this important limitation in the final conclusion.
Reviewer 2 Report
I am satisfied with the revision and suggest the acceptance of the manuscript.
